# Multi-Objective Learning for Deformable Image Registration

**Monika Grewal**[1]                                                MONIKA.GREWAL@CWI.NL
**Henrike Westerveld**[2]                                        G.WESTERVELD@ERASMUSMC.NL
**Peter A. N. Bosman**[1,3]                                           PETER.BOSMAN@CWI.NL
**Tanja Alderliesten**[4]                                        T.ALDERLIESTEN@LUMC.NL

[1] *Centrum Wiskunde & Informatica, 1098 XG, Amsterdam, The Netherlands*

[2] *Erasmus University Medical Center, 3015 GD, Rotterdam, The Netherlands*

[3] *Delft University of Technology, 2628 CD, Delft, The Netherlands*

[4] *Leiden University Medical Center, 2333 ZC, Leiden, The Netherlands*

**Editors:** Accepted for publication at MIDL 2024

## Abstract

Deformable image registration (DIR) involves optimization of multiple conflicting objectives, however, not many existing DIR algorithms are multi-objective (MO). Further, while there has been progress in the design of deep learning algorithms for DIR, there is no work in the direction of MO DIR using deep learning. In this paper, we fill this gap by combining a recently proposed approach for MO training of neural networks with a well-known deep neural network for DIR and create a deep learning based MO DIR approach. We evaluate the proposed approach for DIR of pelvic magnetic resonance imaging (MRI) scans. We experimentally demonstrate that the proposed MO DIR approach – providing multiple DIR outputs for each patient that each correspond to a different trade-off between the objectives – has additional desirable properties from a clinical use point-of-view as compared to providing a single DIR output. The experiments also show that the proposed MO DIR approach provides a better spread of DIR outputs across the entire trade-off front than simply training multiple neural networks with weights for each objective sampled from a grid of possible values.

**Keywords:** Deformable Image Registration, Deep Learning, Multi-objective Optimization, Multi-objective Learning

## 1. Introduction

Deformable image registration (DIR) refers to the task of finding a non-linear transformation that aligns two images. The non-linear transformation is characterized by a deformation vector field (DVF), that maps each location in the target image (also referred to as fixed or reference image) to a location in the source image (also referred to as moving image). The source image is then warped by resampling from the mapped locations. Some of the potential applications of DIR in medical imaging are dose accumulation in radiation treatment, contour propagation, tumor growth tracking, and creating a digital atlas (Mohammadi et al., 2019; Rigaud et al., 2019; Zhao et al., 2022; Salehi et al., 2022).

DIR involves optimization of a parameterized DVF to maximize the similarity between two images. However, optimizing only for maximizing image similarity may yield a highly irregular or sometimes physically implausible DVF due to model overfitting. Therefore, an additional objective penalizing irregularity in the DVF is often used, which inherently

conflicts with the objective of maximizing image similarity (Li and Fan, 2018; Balakrishnan et al., 2019; De Vos et al., 2019). Further, an additional guidance objective (either maximizing the similarity between organ contours or minimizing the distance between corresponding landmarks) is often utilized in challenging DIR problems (Balakrishnan et al., 2019; Hering et al., 2021). Intuitively, improvement in the additional guidance objective should always lead to improvement in the image similarity objective. However, in practice, the additional guidance objective may still conflict with the image similarity objective. This is often caused when the optimization gets overfitted to the regions where additional guidance is provided, deteriorating performance in other image regions (Balakrishnan et al., 2019). Another cause for conflict between the image similarity objective with the additional guidance can be the uncertainty in the additional guidance, which, in turn, could be caused by either inter/intra-observer variance in case of manual annotation or modeling error in case of automatic generation of additional guidance. Therefore, DIR is essentially a multi-objective (MO) problem (Deb et al., 2016), which involves two or more conflicting objectives. This implies that fundamentally an MO approach is appropriate for DIR, where multiple DIR outputs corresponding to a diverse range of trade-offs between the conflicting objectives are provided to the clinicians to a posteriori choose the best solution. Although the notion of DIR being multi-objective is well accepted and discussed, not many DIR approaches have been developed with this perspective. Alderliesten et al. (2015) provided a proof-of-concept study for MO DIR of 2D images. Pirpinia et al. (2017) used an evolutionary algorithm to tune the corresponding weights of different objectives for each 3D breast MRI pair and run single objective DIR multiple times. Nakane et al. (2022) formulated DIR as MO problem by partitioning the template image into several overlapping regions. Andreadis et al. (2023) presented the first integral approach to MO DIR that could be used for 3D volumetric scans using an MO optimization algorithm.

With the advent of deep learning in the past few years, multiple deep learning based DIR approaches have been proposed (Balakrishnan et al., 2019; de Vos et al., 2017; Li and Fan, 2018; Li et al., 2022; Salehi et al., 2022; Rigaud et al., 2019), which provide the possibility to predict the DVF for an entire volumetric scan within seconds. However, to the best of our knowledge, there is no work done in the direction of MO DIR using deep learning. In this paper, we fill this gap and provide a novel approach for MO DIR using deep learning. To this end, we employed a well-known deep neural network for DIR, VoxelMorph (Balakrishnan et al., 2019), and combined it with a recently proposed technique for training neural networks multi-objectively (Deist et al., 2023). Our main contributions are the following:

- We develop a deep learning based approach for MO DIR so that multiple DIR outputs corresponding to different trade-offs between multiple objectives can be presented to the clinical experts for a posteriori decision-making (Hwang and Masud, 2012).
- We demonstrate MO DIR for a challenging real-world registration task: DIR of female pelvic magnetic resonance imaging (MRI) scans and highlight its potential benefits.

## 2. Approach

We first provide a brief background on the concepts of MO optimization that we apply to deep learning based DIR. MO optimization refers to minimizing[1] a vector of $n$ objectives

---

1. In this paper, we assume minimization as objectives correspond to losses in deep learning.

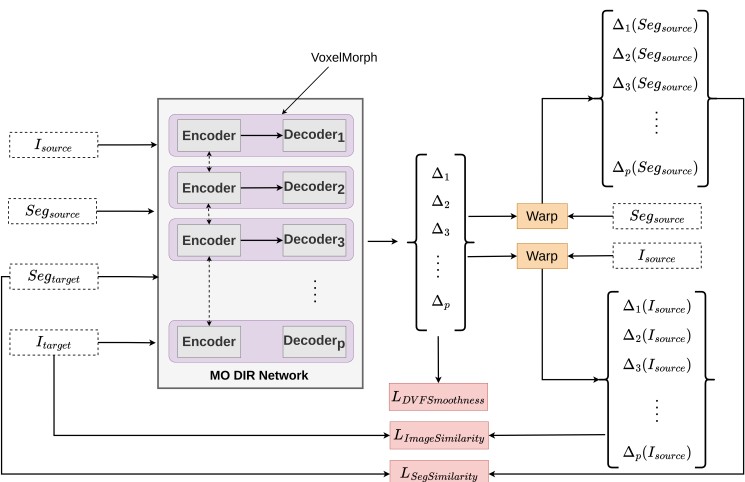

Figure 1: Illustration of the proposed deep learning based MO DIR approach. $I_{source}$: source image, $I_{target}$: target image, $Seg_{source}$ and $Seg_{target}$: organ segmentation masks for source and target image, respectively. The weights of the encoder are shared among $p$ DIR networks, which output $p$ DVFs ($\Delta_1$, $\Delta_2$, ..., $\Delta_p$) to warp $I_{source}$ and $Seg_{source}$. The network is trained to simultaneously minimize $p$ loss vectors $[L_{ImageSimilarity}, L_{DVFSmoothness}, L_{SegSimilarity}]$ using MO learning.

simultaneously. The goal is to find a set (often referred to as 'approximation set') of $p$ solutions that are both close to as well as diversely-spread along the Pareto front – the set of all Pareto optimal solutions in objective space. A solution is Pareto optimal if none of the objectives can be improved without a simultaneous detriment in performance in at least one of the other objectives (Van Veldhuizen and Lamont, 2000).

Our deep learning based MO DIR implementation consists of a DIR network within the MO learning framework proposed in Deist et al. (2023). We selected VoxelMorph (Balakrishnan et al., 2019) for DIR because it is a well-known neural network for DIR. VoxelMorph uses an encoder-decoder style neural network for predicting a DVF, which is a basis for many deep learning based DIR approaches proposed afterwards. We selected the MO learning framework proposed in Deist et al. (2023) for two reasons: a) it achieves MO training of neural networks through hypervolume (HV) maximization - a process that inherently ensures Pareto optimality[2] and diversity between the solutions, b) it is the only MO approach that allows training neural networks multi-objectively without a priori knowledge of the exact preference between different objectives. It should be noted that the latter is crucial in the task of DIR. This is because earlier literature suggests that the exact preference between different objectives may be different between different image pairs, which may only be known a posteriori after inspecting multiple solutions (Pirpinia et al., 2017).

In this paper, we aim to minimize $p$ loss vectors (corresponding to $p$ solutions or DIR outputs in the approximation set), each comprising of three losses: $L_{ImageSimilarity}$, $L_{DVFSmoothness}$, and $L_{SegSimilarity}$. Here, for $L_{ImageSimilarity}$, we used normalized cross-correlation loss. $L_{DVFSmoothness}$ is the squared sum of spatial gradients of the predicted DVF in all directions, and $L_{SegSimilarity}$ is the Dice loss between the fixed image's organ mask

---

2. If HV is maximal, all the solutions are Pareto optimal.

and the moving image's organ mask warped by the predicted DVF (refer to Balakrishnan et al. (2019) for details). In the original formulation of MO learning in Deist et al. (2023), $p$ neural networks are required corresponding to $p$ solutions in the approximation set. Due to the memory intensive nature of training a 3D DIR network, this poses a challenge due to limited GPU memory. To tackle this, we modified the original implementation by sharing the weights of the encoder between $p$ DIR networks as shown in Figure 1. The DIR network predicts $p$ DIR outputs (DVFs). This is followed by calculation of $p$ loss vectors, which are used in the MO learning framework. The parameters of the DIR network are updated using a dynamic loss formulation, that, for each DIR output is defined as:

$$L^i = w_1^i L_{ImageSimilarity} + w_2^i L_{DVFSmoothness} + w_3^i L_{SegSimilarity} \quad \forall i \in \{1, \ldots, p\} \quad (1)$$

Where, the weights $w_1^i, w_2^i, w_3^i$ are calculated in each iteration using HV maximization described in Deist et al. (2023). This ensures that at the end of the training the DIR outputs (that are used to calculate the $p$ loss vectors) are close to, and diversely distributed along the Pareto front of the three objectives.

MO DIR as described above can be understood as training $p$ DIR networks simultaneously, each with different weights for the loss terms, and the weights being selected automatically such that the HV is maximal. That said, MO DIR is fundamentally different from the traditional single DIR following hyperparameter search for the loss weights. In the traditional set up, the selection of a weight (which translate to a trade-off on the approximation front) for each loss is done a priori based on quantitative comparison of a single aggregated (on a validation set) performance metric. Whereas in MO DIR, the selection is done a posteriori by clinical experts based on qualitative evaluation of multiple criteria specific to each patient.

## 2.1. Data

We retrospectively used data from cervical cancer patients who received brachytherapy treatment at Leiden University Medical Center (LUMC), The Netherlands. We received 136 MRI scan pairs (along with associated contours generated for clinical use of four organs at risk: bladder, bowel bag, rectum, and sigmoid) corresponding to two fractions of brachytherapy treatment in anonymized form after approval from the medical ethics committee. The original resolution of the MRI scans was 0.5 mm × 0.5 mm × 4 mm. We resampled the MRI scans to isotropic voxel spacing of 1 mm × 1 mm × 1 mm because the convolution kernels, downsampling, and upsampling operations in VoxelMorph are symmetric. We used randomly cropped patches of size 192 × 192 × 32 as an input to the neural network. We separated the scans at patient level based on their chronological order of acquisition into train and validation (126 scan pairs), and test (10 scan pairs) splits. On the test scans, a radiation therapy technologist annotated 23 anatomical landmarks (details in Appendix B), which were selected by a radiation oncologist on the basis of their importance in brachytherapy treatment for cervical cancer patients. The placement of landmarks was cross-checked by another radiation oncologist.

## 3. Experiments and Results

We implemented[3] our proposed approach using Python and PyTorch. The training hyperparameters were: number of solutions $p = 27$, initialization = Kaiming He, optimizer =

---

3. The implementation is available at https://github.com/monikagrewal/DL-MODIR/tree/public.

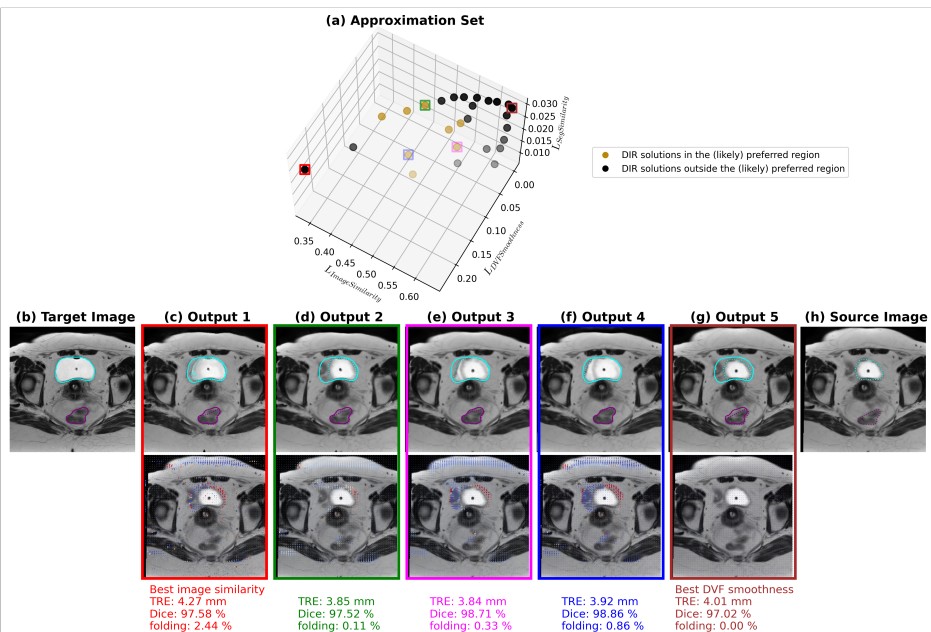

Figure 2: (a) Approximation set. (b) and (h): A transverse slice from the target and source image, respectively. (c)-(g) top row: Warped source images corresponding to five solutions (highlighted with matching frame color) in the set with bladder and rectum contours in cyan and magenta colors, respectively. Solid contours represent the contour in the target image and dashed contours represent the warped source image contour. (c)-(g) bottom row: DVFs overlaid on the source image. Displacement in the x-y plane is represented by direction and scale, and in the z-direction by color (red for cranial, and blue for caudal motion) of arrows.

Adam, learning rate (lr) = $1e^{-4}$, number of training iterations = 20K, reference point for HV calculation = (1, 1, 1) (details in Appendix C). For each experimental setting, we trained 5 models, each corresponding to a different data split. We report their performance on the test set without model selection. To assess the DIR performance, we calculated target registration errors (TREs) of the 23 manually annotated landmarks by transforming the landmarks in the target image with the predicted DVF and calculating the Euclidean distance with the corresponding landmarks in the source image. We also calculated the percentage of voxels with a negative determinant of the spatial Jacobian of the DVF, as an indication of folding in the transformation.

## 3.1. Comparison of MO DIR with Single DIR Output

Contrary to traditional DIR, in MO DIR, the decision maker (in our case a clinical expert) is provided with multiple DIR solutions spread across a range of trade-offs between conflicting objectives. This is demonstrated in Figure 2 (a). The figure shows that there are multiple possible ways to align the two images. In DIR, the solutions at the extremes of the approximation set are likely not interesting because they might be overfitted to a single objective and consequently may yield sub-optimal performance in other objectives. For example, the solution highlighted in the red frame (Output 1) corresponds to minimum

$L_{ImageSimilarity}$, but maximum $L_{DVFSmoothness}$ causing a lot of folding in the DVF. Similarly, the solution highlighted in brown (Output 5) corresponds to no deformation at all. To assist the a posteriori decision-making, such uninteresting solutions can be filtered out by setting acceptance thresholds on each objective. The region of interest in the objective (loss) space where all the acceptance criteria are met, could be considered the preferred region. In Figure 2 (a), we show this region with arbitrarily selected acceptance thresholds ($L_{ImageSimilarity} < 0.55$, $L_{DVFSmoothness} < 0.1$, and $L_{SegSimilarity} < 0.025$).

Within a preferred region of interest, one solution cannot be selected over another based on quantitative comparison of performance metrics as demonstrated in Figure 2. The solution highlighted in green (Output 2) has minimum folding in the DVF, magenta (Output 3) has minimum mean TRE of landmarks, and blue (Output 4) has maximum Dice similarity between organ masks while other metrics are worse. While Output 2 and Output 3 have less folding in the DVF and smaller mean TRE between landmarks, the warped bladder contours (dashed cyan color) considerably deviate from the target bladder contours (solid cyan color) as compared to Output 4. This is due to MO training of the DIR neural network, which ensures that the obtained DIR solutions are all (close to) Pareto optimal i.e., no solution is better than another in any objective without a simultaneous detriment on other objectives. In such a scenario, the most appropriate DIR output can only be selected after visual inspection of the DIR outputs in the preferred region of interest and considering other clinical criteria. For example, the visual inspection of the DVF from Output 4 may reveal that the folding occurs in regions not relevant for brachytherapy treatment. Further, the alignment of the bladder may be more important than the alignment of some landmarks in other regions. Therefore, a clinical expert may prefer Output 4 over Output 3 despite a larger mismatch between landmarks and more folding in the DVF in this test scan pair. Whereas, in another test scan pair, the characteristics of the DVF may be different and the clinical preference may be reversed. Moreover, it is already known from previous research that the weights, which translate to a given trade-off between objectives on the approximation front and the quantitative value of the performance metrics are different in different scan pairs (Pirpinia et al., 2017). This means that the preferred region of interest corresponds to different solutions in the approximation sets from different scan pairs.

Because multiple solutions are provided with MO DIR that are spread in objective space, the clinical expert can navigate through these solutions and select an appropriate trade-off based on the underlying clinical scenario. In contrast, with traditional single DIR, only one of these solutions is provided to the clinical expert. Therefore, the opportunity to evaluate other possibilities and make an informed decision specifically tuned to each patient is lost.

### 3.1.1. COMPARISON OF COMPUTATIONAL OVERHEAD

In the case of single DIR, a DIR network is trained multiple times with different weight combinations for each loss function following a certain strategy. The weights yielding the best aggregated performance on a validation set are used for final training. In MO DIR, multiple neural networks (in our case a single DIR network with multiple decoders) are trained. Therefore, the training overhead of MO DIR in terms of runtime is similar to that of single DIR. However, in MO DIR, the training is done in parallel, requiring more memory. In our implementation, training for $p = 27$ required ∼39 GB and ∼32 GB without and with

a shared encoder, respectively, as compared to ~3.5 GB required for training a single DIR network.

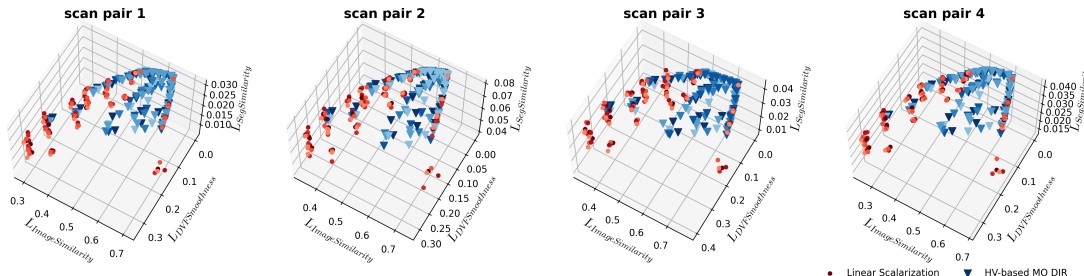

Figure 3: Approximation sets obtained for the first four test scan pairs by linear scalarization (red circles) and the proposed HV-based MO DIR (blue triangles). The approximation sets from five different models trained with different training data splits are shown with slight variations in the color saturation to give an indication of model variance.

## 3.2. Comparison of Proposed MO DIR with Linear Scalarization

In the proposed MO DIR, we used HV maximization to dynamically find the weights for each loss term such that the differently weighted loss training of different neural network heads yields their outputs diversely spread across the approximation front. It may be speculated that a similar diversity of outputs can be trivially obtained by training the different neural networks with uniformly distributed weights for different losses. Such an approach is called 'linear scalarization'. Deist et al. (2023), in their paper, compared linear scalarization with HV maximization for different shapes of Pareto fronts. The authors observed that the translation of the weights to a location on the front is dependent on the shape of the Pareto front, and is as such non-trivial. To investigate this in the case of MO DIR, we compared the proposed HV maximization based MO DIR approach with linear scalarization based MO DIR. To simulate the MO DIR set up with linear scalarization, we trained the different heads of our MO DIR neural network with weights corresponding to diversely distributed points in a grid. We used 27 grid points by enumerating over all the possible combinations for $w_1 \in \{0, 0.5, 1\}$, $w_2 \in \{0, 0.1, 0.5, 1\}$, and $w_3 \in \{0, 0.5, 1\}$ and omitting redundant (e.g., $\{0, 0.5, 0.5\}$ and $\{0.5, 0.5, 0.5\}$). It should be noted that this process of selecting linear scalarization weights is already slightly better than naive linear scalarization.

The approximation sets obtained from linear scalarization vs HV maximization based MO DIR are shown in Figure 3. It is apparent upon visual inspection of the figure that even though the weights used for linear scalarization were diversely distributed, still the obtained solutions are clustered along two edges of the expected triangle-like approximation front. There is a void of solutions in the center region of the expected triangle-like approximation front. This observation corroborates the results in (Deist et al., 2023) - the diverse spread of solutions across the approximation front cannot be obtained trivially through linear scalarization - in the case of DIR as well. In contrast, visual inspection of the solutions in the approximation set obtained using HV maximization based MO DIR, shows a rough triangle-like shape with diversely distributed points in the center as well. This is because HV maximization ensures not only proximity to the Pareto front but also diversity across the approximation front.

## 4. Conclusions and Discussion

We propose the first deep learning approach for MO DIR, which provides multiple DIR solutions diversely spread across the trade-off front between conflicting objectives. With such an approach, clinicians can evaluate multiple DIR solutions that are of potential interest and select the preferred one according to patient-specific and/or treatment-specific clinical criteria. While the prospect of clinicians having to review multiple DIR solutions may seem burdening, in a previous study using a dedicated user interface to navigate MO DIR solutions obtained through optimization (as opposed to deep learning as in this paper), clinicians were positive, considering the use of MO DIR to be insightful (Pirpinia et al., 2018). We also demonstrated that a diverse spread of solutions across the approximation front such as obtained by the proposed MO DIR approach can not be trivially obtained by linear scalarization with diversely distributed weights. Although the potential utility of deep learning based MO DIR is evident from experimental results, the presented work is still only a proof-of-concept. Some of the limitations, open questions, and possible future research directions are as follows:

- HV maximization provides a straightforward way to distribute the solutions diversely on the approximation front without requiring any manual tuning. In future work, it would be interesting to investigate the use of the weighted HV (Zitzler et al., 2007) metric in MO DIR to steer the solutions to a desired region (if such a region can be defined clearly a priori). It is also important to investigate which part of the approximation front is more desired by involving clinicians as a posteriori decision-makers.
- In Figures 2 and 3, the solutions seem more clustered in the region where $L_{ImageSimilarity}$ and $L_{SegSimilarity}$ are large and $L_{DVFSmoothness}$ is small. This could be because solutions in this region of the front are easy to obtain due to no or little deformation, or because of the corresponding shape of and local density along the Pareto front. It is known that setting the reference point differently can impact this (Ishibuchi et al., 2018) (also see Appendix C). It is interesting to investigate this further in the future.
- In our proof-of-principle, we made certain choices e.g., number of objectives, number of solutions in the approximation set, type of additional guidance, type of neural network for DIR, in an effort to create a baseline deep learning based MO DIR approach. That said, the current approach leaves multiple improvement possibilities open in order to realize the complete potential of the MO perspective for DIR. For example, it can be improved by using a more sophisticated neural network for DIR, multi-resolution registration, constraints on tissue types, and diffeomorphism. All of these aspects are independent from the general idea and framework proposed in this paper.
- The presented MO DIR work provides more insights than traditional single DIR approaches by showcasing the trade-offs between different objectives and how these trade-offs differ between scan pairs. However, the objectives are still average values per pair of scans. Practically, the DIR performance will likely not be uniform across the entire scan. Additionally, it is possible that clinically a solution in the vicinity of a provided discrete solution on the approximation front is more desired. It is therefore essential to research in the direction of intuitively visualizing the DVFs and navigating across (and in the local neighborhood of) different solutions.

## Acknowledgments

We thank W. Visser-Groot and S.M. de Boer (Dept. of Radiation Oncology, LUMC, Leiden, NL) for their contributions to this study. This research is part of the research programme Open Technology Programme with project number 15586, which is financed by the Dutch Research Council (NWO), Elekta, and Xomnia. Further, the work is co-funded by the public-private partnership allowance for top consortia for knowledge and innovation (TKIs) from the Dutch Ministry of Economic Affairs.

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

# Appendix A. Effect of Parameter Sharing in the Encoder

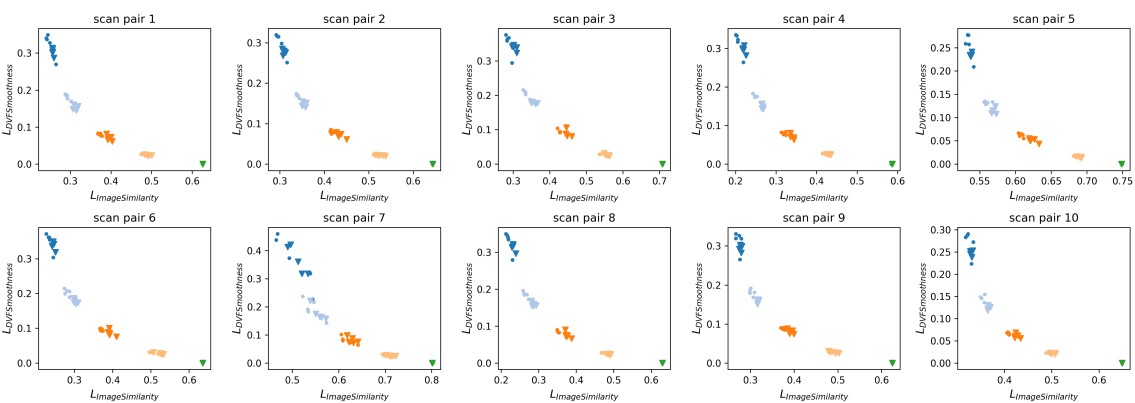

Figure 4: Effect of parameter sharing in the Encoder. filled circles: MO DIR without parameter sharing in the encoder, triangles: MO DIR with parameter sharing in the encoder. $p = 5$, $n = 2$. Approximation sets obtained from 5 models trained on different data splits are shown. Each color represents a DIR solution corresponding to a specific trade-off between $L_{ImageSimilarity}$ and $L_{DVFSmoothness}$.

In Figure 4, 5 approximation sets obtained from 5 models after 5-fold cross-validation, by training the MO DIR approach with $p = 5$ for $L_{ImageSimilarity}$, and $L_{DVFSmoothness}$ losses without (filled circles) and with parameter sharing (triangles) in the encoder are shown for all the test scan pairs. The figure shows that parameter sharing does not impact the distribution of solutions on the front.

## Appendix B.  Description of Landmarks

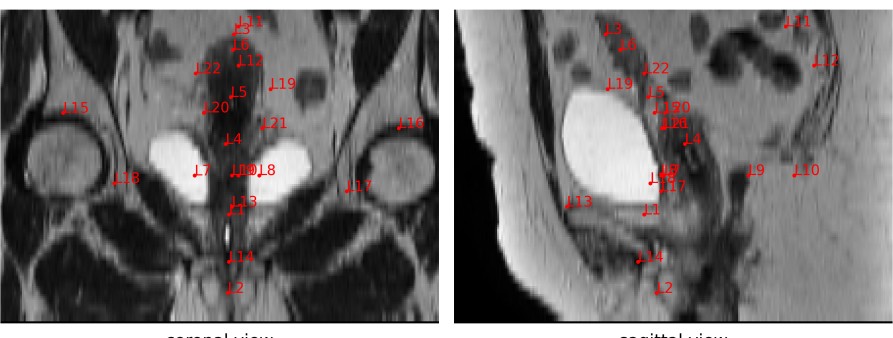

| | coronal view | sagittal view |

| | |
|---|---|
| L1 | Internal urethral ostium |
| L2 | External urethral ostium |
| L3 | Uterus top |
| L4 | Cervical ostium |
| L5 | Isthmus |
| L6 | Intra-uterine canal top |
| L7 | Right ureteral ostium |
| L8 | Left ureteral ostium |
| L9 | Internal anal sfincter |
| L10 | Os coccygis |
| L11 | Most ventral intersections of S2-S3 |
| L12 | Most ventral intersections of S3-S4 |
| L13 | Anterior superior border sympysis (ASBS) |
| L14 | Posterior inferior border sympysis (PIBS) |
| L15 | Right femur head |
| L16 | Left femur head |
| L17 | Left acetabulum |
| L18 | Right acetabulum |
| L19 | Left ligament rotundum |
| L20 | Right entrance of uterine artery to cervix |
| L21 | Left entrance of uterine artery to cervix |
| L22 | Right ligament rotundum |
| L23 | Most ventral intersections of S1-S2 |

Figure 5: Description of landmarks. The landmarks are projected on a coronal (left) and sagittal (right) slice. L23 is not visible in this scan.

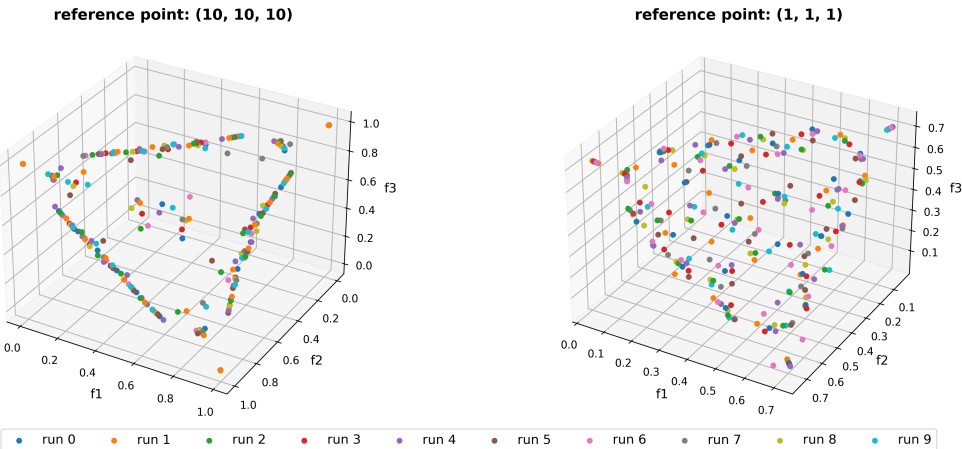

Figure 6: Effect of the location of reference point on the GenMED (Bosman, 2011) benchmark problem. The Pareto front was approximated using 25 points. The solutions from 10 runs are shown for two different locations of the reference point.

## Appendix C. Effect of Selecting Reference Point

The calculation of the HV (and consequently its gradients) is sensitive to the choice of the reference point (Ishibuchi et al., 2018), which, in turn, affects the spread of the solutions on the front. This is particularly the case for three or more objectives. In Figure 6, this phenomenon is illustrated with experiments on the convex GenMED problem with three objectives (Bosman, 2011). Briefly, in the GenMED problem, the $n$ objectives (in our case, $n = 3$ i.e., f1, f2, f3 are the sum of square distances from $n$ unit vectors. When the reference point is far away, the final solutions tend to cluster on the edges of the Pareto front. The spread of the points becomes more uniform across the Pareto front when the reference point is moved closer. Based on these empirical observations, we tuned the reference point for MO DIR training. We considered the following choices: (10, 10, 10), (1, 1, 1), (1, 1, 0.2), (0.5, 1, 1) based on observing the worst loss values after training. For experiments in the paper, we selected (1, 1, 1) as the reference point because it provided well distributed points across the front based on visual inspection on validation set.

## Appendix D. Quantitative Comparison of DIR Performance

Although TRE is a sparse metric and affected by inter- and intra-observer variation in the placement of landmarks, it is often used to quantitatively assess the performance of a DIR method. In this section, we compare the linear scalarization and proposed MO DIR approach described in section 3.2 in terms of mean TRE of 23 landmarks. First, we automatically select a single DIR solution from each approximation set. For this, we assume that a clinical expert would a posteriori select the DIR solution corresponding to minimum mean TRE of 23 landmarks. The underlying idea is that even if the TRE is not explicitly computed, the expert intuitively looks for solutions where landmarks that they are familiar with are well-aligned. In Table 1, we report the mean and standard deviation of this TRE value from 5 models, each trained on a different training data split to provide an estimate of model variance. We

also report the associated folding in the DVF of the selected DIR solution. Although it is difficult to derive any clinical conclusions without inspecting the underlying DVFs, it can be observed that both linear scalarization and HV based MO DIR find quantitatively similar trade-offs between the best TRE values and associated DVF folding. This is not entirely surprising, given that the underlying DL architecture for DIR is the same for both methods.

One might notice a trend of higher TRE values and lower image folding in the selected solutions from HV maximization based MO DIR. However, it is important to realize that the training approach may play a role in this and that training for MO DIR and linear scalarization proceeds differently. Training neural networks with HV maximization is more complex as compared to using fixed weights as in the case with linear scalarization. This is because of the dynamically changing gradients for each network head as a consequence of the HV maximization goal. Therefore, if the exact weights corresponding to the desired trade-off between each objective are known a priori, linear scalarization may yield non-dominated solutions faster. For a fair comparison, we trained the networks in both the linear scalarization and the MO DIR approach with the same number of iterations. It may be possible that this was not the saturation point for both procedures. Ideally, upon saturation, we would expect both linear scalarization and HV maximization to obtain solutions with the same proximity to the Pareto front. However, obtaining the same diversity of solutions (for a given $p$) along the front is not guaranteed for linear scalarization. As demonstrated in section 3.2, this is because the translation from scalarization weights to a well distributed set of solutions along the approximation front is not trivial. Therefore, achieving a diverse spread of solution through linear scalarization would require trying many more combinations. On the other hand, with the HV maximization based MO DIR approach, it can be achieved in a single go.

Table 1: Mean TRE and associated % folding in DVF of the 'best' solution in the approximation set obtained by linear scalarization and MO DIR, respectively for each test scan pair. In each approximation set, the solution corresponding to minimum mean TRE of 23 landmarks is assumed 'best' for the sake of quantitative comparison. Mean ± standard deviation from 5 models trained on different training data splits is reported without model selection.

| Test scan | TRE before | Linear Scalarization | | MO DIR | |
|---|---|---|---|---|---|
| | | TRE | % folding | TRE | % folding |
| 1 | 3.97 | 3.63 ± 0.04 | 0.29 ± 0.19 | 3.74 ± 0.03, | 0.05 ± 0.03 |
| 2 | 4.71 | 4.53 ± 0.11 | 3.45 ± 0.38 | 4.66 ± 0.07, | 2.00 ± 1.23 |
| 3 | 8.21 | 8.04 ± 0.10 | 1.33 ± 1.18 | 8.12 ± 0.06, | 1.07 ± 1.64 |
| 4 | 9.07 | 8.18 ± 0.07 | 0.12 ± 0.15 | 8.58 ± 0.17, | 0.47 ± 0.39 |
| 5 | 4.46 | 4.01 ± 0.06 | 0.80 ± 0.96 | 4.08 ± 0.07, | 1.36 ± 1.01 |
| 6 | 5.55 | 4.52 ± 0.09 | 1.31 ± 0.17 | 4.69 ± 0.09, | 0.76 ± 0.32 |
| 7 | 5.99 | 5.90 ± 0.03 | 0.26 ± 0.18 | 5.93 ± 0.02, | 0.29 ± 0.13 |
| 8 | 4.39 | 3.96 ± 0.05 | 2.72 ± 0.88 | 4.06 ± 0.05, | 1.72 ± 1.31 |
| 9 | 5.73 | 5.06 ± 0.06 | 0.87 ± 0.24 | 5.24 ± 0.13, | 0.82 ± 0.97 |
| 10 | 3.80 | 3.72 ± 0.03 | 0.20 ± 0.28 | 3.70 ± 0.03, | 0.11 ± 0.13 |
| Mean ± SD across patients | 5.59 ± 1.71 | 5.15 ± 1.63 | 1.14 ± 1.21 | 5.28 ± 1.69, | 0.87 ± 1.04 |

