# OpenReview forum: "Multi-Objective Learning for Deformable Image Registration"
_MIDL.io/2024/Conference — MIDL 2024 Poster_

### Official Review · Reviewer_4nYT · 2024-02-26

**Confidence:** 3
**Preliminary Rating:** 2
**Final Rating:** 4

**Summary:**

This paper proposes to combine an approach for multi-objective training of neural networks with image registration. This can be understood as a new way of finding an optimal set of hyper-parameters weighting the different loss terms during the registration.

The authors evaluate their proposed method on 10 MRI scans of the pelvis which includes segmentation of four organs (bladder, bowel bag, rectum, and sigmoid). Furthermore, 23 landmarks per scan were annotated.

The results suggest that the presented MO approach gives comparable results to the baseline but seem not yet to work as hoped.

**Strengths:**

-	The authors addresses the important problem of weighting different objectives during the registration process by using a Multi-objective (MO) strategy. The benefit of this strategy is that multiple solutions are obtained which focus on different objectives. Therefore, after the registration, a clinicians could chose which one fits the best to the application.
-	The paper is well written which makes it easy to follow.
-	The code will be made publicly available after publication.
-	The figures are helpful to understand the paper.

**Weaknesses:**

-	The paper is missing related work in the field of automatic hyper-parameter search for image registration
o	https://arxiv.org/pdf/2101.01035.pdf
o	https://arxiv.org/pdf/2106.12673.pdf
o	Several conventional methods
-	The experimental settings are not clearly reported which makes it hard for the reader to judge the contributions.
-	The TRE does not improve much during the registration suggesting that for both settings (grid search and MO), the registration does not work well (average before 5.588mm, after grid search: 5.164mm, after MO: 5.274mm. The proposed method is even worse than the baseline method. At least if I haven’t done a mistake calculating this, because the authors do not provide the average or median over all cases which makes it even harder for the reader to compare the two methods.
-	 Furthermore, the authors report the “Minimum mean TRE of 23 anatomical landmarks in mm” over 5 different models. This does not make sense since the minimum can be achieved from different methods. (see detailed comments)
-	The authors claim a few things which are not supported by the presented experiments.
-	The discussion is quite limited and should be extended on limitations and failure cases.

**Detailed Comments:**

-	The authors report the “Minimum mean TRE of 23 anatomical landmarks in mm” over 5 different models.
o	The information that cross-validation is performed should be explained in the experiments and not only being mentioned in the table showing the results.
o	Which of the grid-search and MO models is used for this evaluation?
o	Which of the p MO models that is trained per fold of the cross-validation is used? How are those models grouped as ‘similar’.
o	Using the minimum mean TRE could  result in evaluating different models per case. That does not tell something about the general performance of the model.
-	The authors argue that clinicians have the possibly to chose one displacement field (per case) focusing on one of the objectives. However, in the clinical setup, no additional metrics (like TRE) are provided (number of foldings could be provided) resulting in a difficult choice. Often it is not obvious which displacement field is optimal, especially not to a clinician who does not understand the technical background.
-	What does it mean that an algorithm is multi-objective? That they don’t use a MO strategy to weight the objectives?  “not many existing DIR algorithms are multi-objective (MO).”
-	“Traditionally, a set of neural networks are trained with a weighted loss formulation with weights for each loss sampled from a grid of possible values.” Grid search is not a good strategy to find an optimal set of parameters. I recommend to read the following paper https://www.jmlr.org/papers/volume13/bergstra12a/bergstra12a.pdf
-	“Figure 3 shows that the obtained solutions from grid search are clustered in certain regions and do not provide diversity in trade-offs between objectives. This demonstrates the difficulty in tuning the weights of different objectives to achieve a diverse range of trade-offs with grid search.” This conclusion cannot be drawn from the figure and the present experiment. It may also be that the choice of weights (only 3 possibilities, one of which is 0) is not a good choice, which leads to this happening.
-	Are the different loss terms normalized to [0,1] or in which range are those? Does it make sense to have them equaly weighted?
-	“We also demonstrated that the MO DIR setting is more efficient than a grid search for weights for each objective in terms of spread of solutions across the approximation front.” This conclusion cannot be drawn from the presented results. Please be carful in claiming certain things without having proof for it. Already rephrasing the sentence could get the appropriate nuance in.
-	A figure of the positions of the landmarks would be helpful.
-	In Figure 4, what does the colors encode?
-	Figure 5 is not clear to make. Please add an more detailed description.

**Justification Of Final Rating:**

After reviewing the rebuttal, I have decided to adjust my rating to a weak accept. The quality of the paper has been enhanced through the authors' efforts to discuss and address my concerns. Given the improvements and the intriguing nature of the work, I believe this paper merits further discussion at the MIDL conference.

**Justification Of The Preliminary Rating:**

The paper is proposes an interesting approach addressing the hyper-parameter search for image registration. The method does not yet appear to be fully developed, which is why no successes have yet been achieved. This is completely ok. A proof-of-concept paper can also be accepted at the MIDL. However, the results should be presented and discussed in an fair and transparent manner. This is not quite the case yet, but can be addressed in the rebuttal.

**Questions To Address In The Rebuttal:**

- Please improve the description of the experiments such that it is clear.
- Please rework the presentation of the results and provide not the minimum over 5 models but the average and explain how they are chosen.
- Please add the average of over all 10 cases. Currently the grid-search method achieves better results which are not clearly visible to the reader because first the average has to be computed.
- Discuss the results in a fair and open way. What can the reader still learn from this paper? What are limitations?

---

> ### Author Response · Authors · 2024-03-16
> **Thanks for your feedback. Below we provide clarifications for better understanding. We also adapted the paper for clarity of presentation.**
>
> We would like to point out what appears to be a misunderstanding of the paper by the reviewer. The key idea of the paper is to provide an approach to perform deep learning based DIR multi-objectively. The proposed approach is *NOT a way of finding an optimal set of hyper-parameters weighting the different loss terms*. In fact, the premise of the paper is: a) DIR is inherently multi-objective, b) *there is no optimal set of hyper-parameters weighting the different loss terms* that is optimal for each test scans pair (https://pubmed.ncbi.nlm.nih.gov/28436922/).
>
> We made following changes in the paper to provide better understanding and avoid confusion.
>
> - We understand that the MO approach is new to machine/deep learning. Therefore, we have provided a brief background on the MO concepts in section 2 paragraph 1 of the paper. We also added more references.
>
> - We realized that the use of the term `grid search` could have caused confusion that the paper is related to hyperparameter search. To avoid confusion, we have changed it to `linear scalarization`, which is more commonly used in MO literature.
>
> - The main aim of MO DIR is to provide multiple solutions, each corresponding to a different trade-off between objectives, which translates to trade-offs between performance metrics. We argue, and demonstrate how TRE or any other single performance metric is not the full story. There is always a trade-off to make, and the trade-off can only be made after visual inspection of several possible trade-offs, hence the need for multiple outputs instead of one. That said, we did not aim to achieve a better TRE. In fact, we nowhere claim a better performance in terms of improving TRE values. Neither do we create a doubt that the reported TRE values cannot be obtained by traditional methods. It is furthermore important to realize that the maximum possible performance that can be obtained (in terms of TRE) is bound by the underlying DIR DL architecture (VoxelMorph), which is the same for both MO DIR settings. To avoid dilution of attention towards TRE, we have shifted the TRE table to appendix D and focused more on the key message of our work (i.e., benefits of performing DIR multi-objectively) in the main paper. In the appendix, we provide more details regarding reporting of TRE values.
>
> - We made changes in Figures 4 and 5 following the reviewer's suggestions.
>
>
> ### Response to other points
>
> >*The authors claim a few things which are not supported by the presented experiments.*
>
> It is unfortunate that our main message was not clear from the presented experiments. We have modified the presentation of results based on feedback from other reviewers. We are happy to make further changes if the reviewer could be more specific.
>
> >The discussion is quite limited and should be extended on limitations and failure cases.
>
> We agree that the presented discussion in the paper does not cover all the nuances of the field. As stated in the paper, the idea of MO is new to the field of DL (and DL-based DIR) and the presented paper is the first proof-of-principle. Out of 8 pages, one page is dedicated to the discussion section, in which we feel we addressed the most important topics.
>
> >*... Often it is not obvious which displacement field is optimal, especially not to a clinician who does not understand the technical background.*
>
> The reviewer is right to point out that in clinical setup, the metrics like TRE can not be provided, only the information on the objectives considered for learning such as image similarity and deformation magnitude (and foldings) can be provided. We argue that the selection of the appropriate solution is not possible by quantitative comparison, the decision maker does not require technical knowledge, but domain expertise on what matters more in a clinical scenario. For example, does alignment of bladder matter more or alignment of a femur head. The clinicians know the answer to such questions best.
>
> >*What does it mean that an algorithm is multi-objective?...*
>
> In this sentence, we intend to say that not many existing DIR algorithms provide multiple DIR outputs diversely spread across the trade-off front between the conflicting objectives.
>
> > *What can the reader still learn from this paper? What are limitations?*
>
> The main take-home message from the paper are:
>
> a) as was already known, MO DIR provides a possibility to a posteriori select a suitable trade-off between the conflicting objectives, specifically tuned to each test case scenario, but now we can also achieve this through deep learning.
>
> b) in the (deep) learning setting, MO DIR can not be achieved trivially through linear scalarization with uniformly distributed weights.
>
> We have discussed limitations of our work in section 4 as much as the page limit allowed in the paper.

---

> ### Comment · Reviewer_4nYT · 2024-03-21
>
> Thank you for the insightful explanations and the improvements made in the rebuttal.
>
> - I appreciate the clarification regarding the motivation behind MO Learning. I acknowledge the perspective presented, but would like to offer an alternative view where MO Learning could also be understood as a sophisticated form of hyperparameter optimization. This is not a search for a single optimal parameter set, but rather an exploration of a spectrum of viable alternatives from which clinicians can choose, tailored to their specific requirements. The core of my argument lies in the alignment of objectives with the different components of the loss function, each of which can be weighted differently. This flexibility allows for the training of different models that emphasize different aspects such as image similarity or the importance of segmentation masks. By presenting these different models to physicians, they are able to choose the configuration they find most effective. One of the advantages of MO learning, in my view, is its ability to make an "optimal" selection and thus avoid the limitations that might result from a less careful selection of weights.
> With this in mind, I would argue that MO Learning can be understood as an innovative approach to hyperparameter selection.
>
>    I would be grateful for your thoughts on this perspective. In particular, I am curious about your reasons why you see this view as potentially flawed. Please note that in the original review I said that MO can be **understood** as a hyperparametric method, not that it is one :)
>
> - *It is unfortunate that our main message was not clear from the presented experiments. We have modified the presentation of results based on feedback from other reviewers. We are happy to make further changes if the reviewer could be more specific.*
> The points are mentioned in detailed comments of the original review.
>
> - For example, how have you demonstrated that
> “that the MO DIR setting is more efficient than a grid search for weights for each objective in terms of spread of solutions across the approximation front.” ?
>
>    It's conceivable that I may have misinterpreted the experimental design or findings, but the evidence supporting this assertion wasn't immediately apparent to me. Could you please clarify this?
>
> - Lastly, I would kindly request the inclusion of the average TRE for all 10 cases in table 1 to provide a more direct and comprehensible comparison with the grid search method's outcomes.

---

> > ### Author Response · Authors · 2024-03-22
> > **Thanks for the discussion. We made adaptations in the paper and address the points in comment.**
> >
> > > *Comparing MO learning with hyperparameter optimization*
> >
> > We thank the reviewer for elaborating on their perspective. We agree that for some cases, there are clear similarities between hyperparameter search and MO learning. For example, we agree with the reviewer’s sentence `This is not a search for a single optimal parameter set, but rather an exploration of a spectrum of viable alternatives from which clinicians can choose, tailored to their specific requirements.` And the explanation afterward.
> >
> > That said, we see the sentence `One of the advantages of MO learning, in my view, is its ability to make an "optimal" selection and thus avoid the limitations that might result from a less careful selection of weights.` differently. We would rather prefer to say that `MO learning enables the clinicians to make an “optimal” selection based on other clinical criteria that were not part of DIR.` In other words, in MO learning, the decision-maker is the clinical expert , who makes a final selection a posteriori using qualitative metrics. On the other hand, in hyperparameter search, selection is done a priori based on quantitative comparison of a single aggregated (on a validation set) performance metric.
> >
> > > *how have you demonstrated that “that the MO DIR setting is more efficient than a grid search for weights for each objective in terms of spread of solutions across the approximation front.” ?*
> >
> > We would refer the reviewer to section 3.2 and Figure 3 of the updated paper. We have adapted the text for more clarity of presentation.
> >
> > > *inclusion of the average TRE for all 10 cases in table 1*
> >
> > We have adapted Table 1 according to the suggestion. However, we would like to point out that the key motivation behind MO DIR is to provide a DIR solution separately tuned to each patient-specific criterion. With this perspective, an average for all 10 cases is not necessarily the most informative. Moreover, since the underlying registration model is the same for the linear scalarization and the HV maximization based MO DIR approach, it is not necessarily to be expected that the final TRE will be much different.
> >
> > Because TRE values are generally of interest for DIR, we presented the TRE table as a sanity check. With the TRE table in appendix D and experiment in section 3.2, we want to demonstrate that nothing changes in terms of TRE values by doing MO DIR using HV maximization instead of using linear scalarization. The only thing that changes is that we obtain diversity in solutions across the front in a more straightforward manner, which with linear scalarization would require trying many more combinations.
> >
> > --------------------------------------------------------------
> >
> > We apologize that we missed addressing the following two points from the detailed comments section in the original review. We address them in the updated version.
> >
> > > *It may also be that the choice of weights (only 3 possibilities, one of which is 0) is not a good choice, which leads to this happening.*
> >
> > We want to point out that the number of possibilities were chosen such that the total number of combinations after omitting redundant combinations is equivalent to $p$ (27) used for HV based MO DIR for fair comparison. $p=27$ was chosen keeping in mind that these many points would give a sufficient impression of an approximation of a trade-off front for 3 objectives. Further, choosing one of the possibilities for a weight as 0 is essential to know the extremes of the approximation set.
> >
> > > *Are the different loss terms normalized to [0,1] or in which range are those? Does it make sense to have them equaly weighted?*
> >
> > The $L_{ImageSimilarity}$ and $L_{SegSimilarity}$ are bound to [0, 1]. The lower bound on $L_{DVFSmoothness}$ is 0. In our experiments, we observed that the upper range of $L_{DVFSmoothness}$ was ~0.4.
> >
> > We performed some preliminary experiments on normalizing the loss terms with a motivation to bias the spread of solutions across the approximation front. We do not have any conclusive results to share from those experiments. The main reason is that the translation of normalization to a point on the approximation front is affected by the shape of the Pareto front and optimization landscape. It will be interesting to investigate this further in the future.
> >
> > We interpret that by *equally weighting the losses*, the reviewer intends to say *giving equal importance to each loss*. Please correct, if otherwise. We have discussed this in the first bullet point in section 4. We also speculate that likely one objective (or a specific region of the approximation front) may be more important than others. We believe that with the use of **weighted HV** (https://link.springer.com/chapter/10.1007/978-3-540-70928-2_64), this can be achieved in the most intuitive way. It will be interesting to investigate this in the future.

---

> > > ### Comment · Reviewer_4nYT · 2024-03-26
> > >
> > > I extend my gratitude to the authors for their comprehensive response, which has provided a clearer understanding of the paper's objectives. While I recognize the challenges posed by the page limit constraints, I recommend that the authors contemplate integrating these detailed explanations into the manuscript itself. Enhancing the clarity of the paper's objectives in this manner would greatly facilitate readers' comprehension and improve the paper's accessibility.
> > >
> > > Still, I would like to inquire about the authors' perspective on a particular aspect of the study: In the multi-objective (MO) training process, specific objectives are employed to construct the Pareto front, guiding the solutions predominantly toward these goals. The authors mention that MO learning allows clinicians to make an "optimal" choice based on additional clinical criteria not included in the direct image registration (DIR) process. My question concerns the extent to which the solutions generated are independent of the predefined objectives.
> > >
> > > To illustrate, if the training only utilizes segmentation masks and aims at objectives like plausibility and alignment with these masks, it's natural to expect that the MO solutions primarily address these factors. However, a clinician might be interested in aligning particular landmarks or structures not considered during the training phase. While it seems that the MO approach can partially accommodate such needs, I believe it would be valuable to explore how the method performs in adapting to criteria beyond the initial training objectives in greater depth.

---

> > > > ### Author Response · Authors · 2024-03-26
> > > >
> > > > We thank the reviewer for their feedback regarding improving the manuscript. Following the reviewer's suggestion, we have added a paragraph in the paper (please refer to page 4, paragraph 2), which provides more details on MO DIR and how MO learning distinguishes from hyperparameter search.
> > > >
> > > > **Regarding MO learning adapting to criteria beyond the initial training objectives**:
> > > >
> > > > The reviewer’s suggestion appears to point in the direction of MO DIR in an adaptive or active learning manner. We have no particular solutions for this at this time. However, we can speculate that an iterative learning process, wherein, the decision makers provide feedback on what additional objective might be of interest after evaluating the results from first MO DIR training, might be possible. Consequently, a new DIR network can be trained using MO learning with the additional objective suggested by the decision-makers.

---

### Official Review · Reviewer_DR7q · 2024-02-28

**Confidence:** 4
**Preliminary Rating:** 3
**Recommendation:** Poster
**Final Rating:** 3.5

**Summary:**

This paper introduces the application of multi-objective optimization for unsupervised image registration, tackling the issue of conflicting objectives encountered during the training of registration networks like Voxelmorph and its variants. Traditionally, these methods strive to maximize visual similarity while ensuring the displacement field remains smooth and invertible. Additional objectives might include matching labels or keypoints, where optimizing for one objective could lead to suboptimal results for others. The authors propose utilizing hypervolume maximization to generate a set of solutions that represent a balanced trade-off among all objectives. This is achieved by modifying the standard Voxelmorph architecture with a shared encoder and separate decoders for each solution. The optimization of each solution targets a weighted sum of intensity similarity loss, label similarity loss, and smoothness loss, with the weights themselves being the focus of hypervolume maximization. The paper evaluates this approach by optimizing across 27 possible outcomes and compares the merits of this method against traditional hyperparameter optimization.

**Strengths:**

The recognition of image registration as a multi-objective problem and the novel application of hypervolume maximization to address this challenge are significant strengths of this work. This approach is innovative and provides a promising direction for handling the inherent trade-offs between different objectives in image registration tasks.

**Weaknesses:**

The paper falls short in detailing the hypervolume optimization process, which is central to the novelty of the proposed method. The description of the network architecture, while necessary, does not contribute significant novelty and overshadows the more critical aspects of the optimization process. A deeper dive into the optimization challenges, including memory demands and computational complexities, would have been valuable.

**Detailed Comments:**

The paper would benefit from a more detailed explanation of the hypervolume maximization process, including its implementation and integration with the modified network architecture. Clarifying these aspects would greatly enhance the paper's contribution and allow for a better understanding of its potential impact.

**Justification Of Final Rating:**

Thank you very much for revising the manuscript and addressing my questions. The paper has noticably improved. However, in my opinion, the paper lacks sufficient novelty so I will remain with my previous decision of 'Borderline'.

**Justification Of The Preliminary Rating:**

While the concept of applying hypervolume maximization to unsupervised image registration is intriguing and presents a novel approach to addressing the multi-objective nature of the problem, the paper lacks sufficient detail on the optimization process itself. Additionally, committing to making the implementation code publicly available upon publication would significantly enhance the paper's value, enabling replication and further exploration of the proposed method.

**Questions To Address In The Rebuttal:**

Could you clarify if an evolutionary algorithm is utilized for the hypervolume maximization?
How does the quality of the "best" solution compare to the "worst," and are there strategies in place to eliminate less promising solutions to reduce the manual review burden of potential candidates?

---

> ### Author Response · Authors · 2024-03-16
> **Thanks for your feedback. We have addressed the comments point-by-point and added more details in the paper.**
>
> ## Weaknesses
>
> >*The paper falls short in detailing the hypervolume optimization process, which is central to the novelty of the proposed method.*
>
> We would like to point out that we did not develop the hypervolume optimization process as a part of this work. We use our previously developed approach ([Deist et. al,](https://dl.acm.org/doi/abs/10.1007/978-3-031-27250-9_8)) to train a DIR neural network in this paper. The main novelty of our paper comes from presenting a proof-of-principle approach to achieve MO DIR using deep learning. We refer the interested reader to Deist et.al, for details on hypervolume optimization.
>
> > *A deeper dive into the optimization challenges, including memory demands and computational complexities, would have been valuable.*
>
> We thank the reviewer for their feedback. We have added section 3.1.1 on comparing the memory and computational overhead of our MO DIR approach compared to the single DIR setting.
>
> ## Detailed Comments
> > *The paper would benefit from a more detailed explanation of the hypervolume maximization process, including its implementation and integration with the modified network architecture.*
>
> We would like to point out again that a detailed explanation of hypervolume maximisation is beyond the scope of this paper. We refer the reviewer to the original work on the use of hypervolume maximization for training neural networks ([Deist et. al,](https://dl.acm.org/doi/abs/10.1007/978-3-031-27250-9_8)) for a detailed understanding. In section 2, paragraph 3, we explain the integration of HV maximization based training with a DIR neural network, which is the main contribution of this paper.
>
> ## Questions To Address In The Rebuttal
>
> >*Could you clarify if an evolutionary algorithm is utilized for the hypervolume maximization?*
>
> We did not use an evolutionary algorithm for hypervolume maximization. We used a gradient based approach, in which, the hypervolume maximization of a set of neural networks can be reformulated as training the neural networks with a dynamically weighted loss. The dynamic weights are the gradients of hypervolume in loss space. We refer to the original article ([Deist et. al,](https://dl.acm.org/doi/abs/10.1007/978-3-031-27250-9_8))  for further details.
>
> >*How does the quality of the "best" solution compare to the "worst,".*
>
> There are no “best” or “worst” solutions that can be decided quantitatively. All the solutions are, in principle, (close to) Pareto optimal, which means no solution is better than another solution in one objective without a simultaneous detriment in other objectives.
> In the paper, we say “best” solution with respect to a specific criterion i.e., TRE, percent folding, or matching of segmentation masks. We have updated the experiment section of our paper. We hope that it presents the results more clearly than before.
>
> > *are there strategies in place to eliminate less promising solutions to reduce the manual review burden of potential candidates?*
>
> We do not have such a method in our paper. However, we can potentially a posteriori eliminate a large section of the approximation front corresponding to solutions at the extremes i.e., the solutions corresponding to very high DVFSmoothness loss, or ImageSimilarity loss, as they are extreme and therefore unlikely to be of interest. We have demonstrated this in Figure 2 of the updated paper.

---

> > ### Comment · Reviewer_DR7q · 2024-03-27
> >
> > Thank you very much for revising the manuscript and addressing my questions. I have no further questions at this time.

---

### Official Review · Reviewer_jMhn · 2024-02-29

**Confidence:** 3
**Preliminary Rating:** 3
**Recommendation:** Poster
**Final Rating:** 3.5

**Summary:**

The authors adapted a deep learning method specifically for multi-objective deformable image registration to medical imaging setting. The authors tested the MO DIR framework with pelvic MRI images annotated for radiation oncology treatment for cervical cancers. The authors compared output of MO DIR vs that of single DIR.

**Strengths:**

The authors focused on a relatively unique application area in DIR that is a rare area of study but important for clinical application of ML.
The authors used well annotated dataset that have anatomic landmark and annotated lesions to train and evaluate the model respectively.

**Weaknesses:**

How MRI scans are sampled and how the images are separated into train/validation/test may impact results. This may warrant additional discussion and/or experiments in addition to what is being said in the paper.

For the results section, why presentation of different solutions matters in this application scenario seems unclear and it seems unclear why MO DIR is better than single DIR.

**Detailed Comments:**

The discussions emphasized added value of MO DIR but this is not apparent in the results.
Choice of patient-level separation of data makes sense to some degree though more explanation of rationale for sampling needs more details.
The direct impact of MO DIR in clinical application in this setting seems a bit unclear at this moment.

**Justification Of Final Rating:**

I appreciate the authors' update on the paper that address most of my questions and suggestions, and their detailed reply to other reviewers as well. I think the updates address some of my technical questions and I think the authors could improve on novelty of their technical approach in future research work.

**Justification Of The Preliminary Rating:**

Overall this project is unique in problem area of choice and used appropriate set of data for application of MO DIR. However, choice of MO DIR framwork, choice of sampling, real-life meaning and significance of the results seem unclear at this moment.

**Questions To Address In The Rebuttal:**

How MRI scans are sampled and how the images are separated into train/validation/test may impact results. This may warrant additional discussion and/or experiments in addition to what is being said in the paper.

For the results section, why presentation of different solutions matters in this application scenario seems unclear and it seems unclear why MO DIR is better than single DIR.

**Special Issue:**

No

---

> ### Author Response · Authors · 2024-03-16
> **Thanks for your feedback. We have addressed the comments point-by-point and added more details in the paper.**
>
> > *How MRI scans are sampled and how the images are separated into train/validation/test may impact results.*
>
> We thank the reviewer for pointing out this shortcoming in the paper presentation. The reviewer is right to speculate that MRI sampling may impact the results because CNN kernels and down/up-sampling techniques are all isotropic. To avoid any bias between processing of transverse vs in-plane direction, we resampled the MRI scans to isotropic voxel spacing. We have added this information in section 2.1 of the updated paper.
>
> Further, the separation of train, validation, and test scans was done as follows. First, the test scans were separated by sorting the scans in chronological order and taking the latest scans as test scans. The remaining scans were split into 5 folds by random sampling. This was followed by training 5 models, each utilising 4 splits for training and the 5th split for early stopping. The mean and standard deviation of the performance of these models on the test data is reported without doing model selection. We have added this information in section 3, first paragraph of the updated paper.
>
> > *For the results section, why presentation of different solutions matters in this application scenario seems unclear and it seems unclear why MO DIR is better than single DIR.*
>
> It is unfortunate that the presented results fall short of conveying the key message to the reader. We have updated the experiments section, and modified Figure 2 for a clearer presentation. Briefly summarized, we want to demonstrate that in DIR different competing objectives are of interest. Especially if the DIR task is challenging, the conflict between the objectives is large and many different solutions (deformation vector fields) exist that are equally good according to these criteria and adhering to a multi-objective notion of not being able to a priori weight the importance of these objectives. For instance, a deformation that achieves a better image similarity objective may have unrealistic deformations. The appropriate selection can only be made through visual inspection of potential DIR solutions by a clinical expert, who can also take into account other clinical criteria that are not part of DIR. Moreover, what is the preferred solution also depends on what you want to use the DIR output for. In single DIR, because only one DIR output corresponding to a pre-selected trade-off is provided to the clinician, the opportunity to make patient specific choices is lost.

---

### Official Review · Reviewer_JarU · 2024-03-01

**Confidence:** 4
**Preliminary Rating:** 3
**Recommendation:** Poster
**Final Rating:** 3.5

**Summary:**

The authors proposed to combine a multi-objective training approach with deep learning-based deformable image registration. Evaluated on a MRI dataset, the proposed method outperforms single registration network and grid search of possible loss weights.

**Strengths:**

The paper innovatively addresses image registration as a multi-objective problem, and the proposed approach provides a better spread of DIR outputs.

Instead of doing grid search, the proposed approach used HV maximization to determine the loss weights during each iteration, which is less tedious and more convenient.

The paper is well-written and clear.

**Weaknesses:**

The comparison of MO DIR with single DIR was only through the spread of the solution set, but wasn't through an implementation with another single DIR.

Though the multiple neural networks share the same encoder, there might still be significant memory consumption increase to train p decoders (e.g. 27 decoders with the hyperparameter they selected). A comparison on this will be of interest.

**Detailed Comments:**

After getting the solution set, the final registration result will be manually selected by the decision maker. Will this make the method less automated and requiring additional intervention?

**Justification Of Final Rating:**

I appreciate that the authors addressed my concerns in the previous cycle, and the paper is in a richer context now. However, I am still a bit concerned about whether the manual selection in the provided solution set will lead to better outcomes in various complex clinical settings. A clinician reader study reflecting the preferences will be of great interest in future work. Also, if the "better similarity" or "better smoothness" can be achieved through a tuneable prompt to further automate the framework. Thus I am increasing the rating to borderline accept but will agree with the majority

**Justification Of The Preliminary Rating:**

The work is innovative in proposing DL-based MO DIR, but with concerns about its performance comparison and if the trade-off will lead to substantial time/memory consumption and additional interventions.

**Questions To Address In The Rebuttal:**

As in the previous sections, it will mainly be a comparison with single instance benchmark and the memory/time consumption.

**Special Issue:**

No

---

> ### Author Response · Authors · 2024-03-16
> **Thanks for your feedback. We have addressed the comments point-by-point and added more details in the paper.**
>
> ## Concerns in Weaknesses
>
> > *The comparison of MO DIR with single DIR wasn't through an implementation with another single DIR.*
>
> We would like to remark that MO DIR with $p$ solutions is equivalent to presenting outputs from $p$ single DIR (each corresponding to a different trade-off between objectives) at once. Therefore, we believe that comparison with a single DIR instance would be redundant as each of the $p$ outputs already represents the output from a single DIR run.
>
> A follow up question may arise as to why we didn't just train a DIR network $p$ times, each time with a different weight for each objective. The answer is that the weight given to an objective during training, does not translate trivially to a trade-off on the front after training. It depends on the shape of the front ([Deist et. al,](https://dl.acm.org/doi/abs/10.1007/978-3-031-27250-9_8)), which is not known for DIR or most of other MO applications a priori. We experimentally investigate this in our paper in section 3.2.
>
> Further, we have updated Figure 2 and section 3.1 of the paper to present the comparison of MO DIR with single DIR more clearly.
>
> > *there might still be significant memory consumption*
>
> We thank the reviewer for their feedback. We have added a paragraph (section 3.1.1)  to the paper to compare the memory overhead of training a MO DIR approach as compared to a single DIR approach. Briefly, training the proposed MO DIR with p=27 requires ~32GB GPU RAM as compared to ~3.5GB for training a single DIR, which is significantly more but in our opinion well within the limits of modern GPUs used for training deep neural networks.
>
> ## Detailed Comments
>
> > *After getting the solution set, the final registration result will be manually selected by the decision maker. Will this make the method less automated and requiring additional intervention?*
>
> Yes, from this perspective, MO DIR is indeed a less automated approach than single DIR, which is intentional. The key motivation of MO DIR is to have better potential of adoption in challenging DIR scenarios where the trade-off between the conflicting objectives can not be known a priori. Moreover, the desired trade-off between conflicting objectives is different between patients and may be dependent on clinical criteria not considered during DIR. In such DIR scenarios, the above mentioned reasons necessitate a posteriori evaluation of multiple DIR solutions and manual intervention for decision-making. We have laid out this motivation in the introduction section of the paper. Further, the task of DIR in the pelvic region, which has been selected for experiments in this paper, is an example of such a challenging DIR scenario.
>
> We would also refer to a previous study that provided a dedicated user interface to navigate through the resulting solutions from MO DIR (https://pubmed.ncbi.nlm.nih.gov/30840735/). In this study the clinicians considered MO DIR insightful. We have added discussion on this in section 4, paragraph 1 of the updated paper.

---

### Author Response · Authors · 2024-03-16
**General response to reviewers**

We thank the reviewers for taking the time to review our work and provide feedback. We also thank the reviewers for noticing the novelty and appreciating the innovativeness of the approach, and presentation of the paper. We have made adaptations to our paper to address the points raised by reviewers. Below we provide a point-by-point response to the reviewers’ feedback.

---

### Meta-Review · Area_Chair_bkmN · 2024-04-04

**Recommendation:** Accept (Poster)
**Confidence:** 5

**Metareview:**

Reviewers’ questions were mainly about details on experiments and were well addressed in the authors’ feedback. Reviewers acknowledged that the paper had been improved during the discussion, and their questions were answered.

---

### Decision · Program_Chairs · 2024-04-06

Accept (Poster)